# Multifunctional Opioid-Derived Hybrids in Neuropathic Pain: Preclinical Evidence, Ideas and Challenges

**DOI:** 10.3390/molecules25235520

**Published:** 2020-11-25

**Authors:** Joanna Starnowska-Sokół, Barbara Przewłocka

**Affiliations:** 1Department of Pain Pharmacology, Maj Institute of Pharmacology, Polish Academy of Sciences, 31-343 Krakow, Poland; joanna.starnowska@uj.edu.pl; 2Department of Pharmacobiology, Faculty of Pharmacy, Jagiellonian University Medical College, 30-688 Krakow, Poland

**Keywords:** neuropathic pain, opioids, hybrid compounds, bifunctional designed ligands, nerve injury

## Abstract

When the first- and second-line therapeutics used to treat neuropathic pain (NP) fail to induce efficient analgesia—which is estimated to relate to more than half of the patients—opioid drugs are prescribed. Still, the pathological changes following the nerve tissue injury, i.a. pronociceptive neuropeptide systems activation, oppose the analgesic effects of opiates, enforcing the use of relatively high therapeutic doses in order to obtain satisfying pain relief. In parallel, the repeated use of opioid agonists is associated with burdensome adverse effects due to compensatory mechanisms that arise thereafter. Rational design of hybrid drugs, in which opioid ligands are combined with other pharmacophores that block the antiopioid action of pronociceptive systems, delivers the opportunity to ameliorate the NP-oriented opioid treatment via addressing neuropathological mechanisms shared both by NP and repeated exposition to opioids. Therewith, the new dually acting drugs, tailored for the specificity of NP, can gain in efficacy under nerve injury conditions and have an improved safety profile as compared to selective opioid agonists. The current review presents the latest ideas on opioid-comprising hybrid drugs designed to treat painful neuropathy, with focus on their biological action, as well as limitations and challenges related to this therapeutic approach.

## 1. Introduction

### 1.1. Neuropathic Pain (NP) Differs from Physiological Nociception

When a noxious stimulus—let it be mechanical impact, excessive heat/cold, or exposure to harmful chemicals—is perceived, an unpleasant sensory and emotional experience arises. This universal sensation, known as pain, alerts the individual to potential or actual tissue damage and helps modify behavior to avoid the danger in the future; hence it bears vital evolutionary functions [1]. As opposed to physiological nociception, chronic pain with a neuropathic component (i.e., associated with a lesion or disease of the somatosensory nervous system [2]) is marked by a low correlation between observable injury and nociceptive responses, lasts beyond tissue healing time, and, as such, serves no evolutionary purpose, while exerting a profound negative impact on the quality of life [3]. Worse still, complex alterations in neural processing following the nerve tissue injury make this type of pain less responsive to conventional treatments than acute and inflammatory pain [4,5].

### 1.2. The Efficacy of Currently Available Neuropathic Pain Therapies is Limited

It is assessed that more than half of the neuropathic pain patients do not experience satisfying pharmacotherapy-induced pain relief even to a moderate extent [6,7]. 

First-line pharmacotherapy of NP includes gabapentoids, tricyclic antidepressants, and serotonin-norepinephrine reuptake inhibitors. When the response to these drugs is poor, lidocaine plasters, capsaicine patches and dually acting µ-opioid peptide (MOP) receptor agonists that concurrently inhibit the reuptake of serotonin and/or norepinephrine (tramadol, tapentadol) are to be used as a second-line therapy. Combinations of first-line medications are also recommended at this stage of treatment [8,9].

Opioid drugs (morphine, oxycodone) are prescribed only as the third-line therapy, as they carry the risk of several negative adverse effects when administered in higher doses exacted by neuropathic pain severity. Importantly, physicians report that not all NP patients respond to exogenous opioids equally, and that some of the patients require high doses of synthetic opiates to achieve optimum analgesia, which further enhances the risk of the side effects occurrence [10]. The most common adverse events in opioid therapy include troublesome gastrointestinal motility disturbances and sedation. Besides, the repeated use of opioids—inevitable in chronic pain conditions—is associated with the development of physical dependence and tolerance to the analgesic effect of the initial dose, while overdosing can lead to life-threatening respiratory depression [5,11].

Still, opioids can efficiently alleviate moderate to severe pain of neuropathic origin [12,13], and are used in the clinical practice: patients with NP are significantly more likely to receive opioids than those suffering from chronic pain without neuropathic component [14] due to the limited efficacy of other drugs; it is estimated that less than 35% of NP patients actually benefit from nonopioid treatments [15]. Having this in mind, it is a vital task to make use of the analgesic properties of opioids in NP treatment while minimizing unwanted side effects. Currently available strategies to avoid adverse events related with the long-term therapy (tolerance development, opioid-induced hyperlagesia) involve opioid rotation and the use of adjuvant analgesics [16]. Alternatively, new structures aiming at opioid receptors are designed to make these pharmaceutics better adjusted to the specificity of neuropathic pain mechanisms (which, in turn, could allow to obtain maximal analgesia at the lowest opiate dose possible) and long-term therapy. The successful outcome of new drugs development draws upon the comprehensive understanding of neuropathic pain mechanisms on molecular and systemic level.

### 1.3. Neuropathic Pain Mechanisms Negatively Influence the Opioid-Induced Analgesia

The opioid system, with its three seven-transmembrane-spanning (7TM) G-protein coupled (Gαi–Gαo) µ-, δ- and κ-opioid peptide (MOP, DOP and KOP, respectively) receptors as well as their analgesia-inducing endogenous ligands (β-endorphin, met- and leu-enkephalins, dynorphins), is intrinsically linked to pain processing in mammals, next to the profound role it plays in mood regulation, reward and addiction. Opioid receptors are present both in peripheral tissues and the central nervous system CNS, inhibiting pain transmission at different levels of the ascending and descending pain pathways [1,17]. The opioid receptors are besides expressed in immune cells (lymphocytes, granulocytes, monocytes, macrophages) that can as well release opioid peptides ameliorating pain under pathological conditions [18].

As mentioned earlier, the response to opioid treatment differs among NP patients [10]. Individual differences in opioid responsiveness may be partially explained with genetic polymorphisms on the MOP-encoding Oprm1 gene, as well as on vast families of pharmacokinetics-influencing genes encoding the enzymes that determine the opioid metabolism, basically cytochrome P450 family and the uridine diphosphoglucuronysyltransferases [19]. However, the poor responsiveness to opioids under neuropathic pain conditions is greatly dependent on the specific pathological changes that steadily afflict the nervous system after nerve tissue injury.

Endogenous opioid system serves an adaptive role for the organism by inducing analgesia when noxious environmental stimuli occur, as the ability to temporarily ignore the pain of the injury heightens the chances of surviving a dangerous event [20]. This state is reflected in animal NP models by the increased level of endogenous opioids in pain-facilitating spinal and brain areas, where the opioid peptides can induce their inhibitory effects [21,22]. Nevertheless, as the environmentally produced analgesia must be timely suppressed to help modify the behavior once the danger ceases, continued opioid signaling activated in the presence of nerve tissue trauma initiates the homeostatic compensatory mechanisms that eventually counteract the inhibitory effects of opioid activation [23]. Such a duality is observed within the opioid system itself: for example, an injury to a nerve induces the upregulated expression of an endogenous peptide, dynorphin A(1–17), which can exert neuroinhibitory (antinociceptive) opioid effects, but can as well potentiate allodynia via *N*-methyl-D-aspartate (NMDA) receptors [24] or be cleaved into a nonopioid proteolytic fragment (2–17) that activates excitatory bradykinin receptors and thus promotes pronociceptive effects [25,26,27]. What is more, the elevated expression of an opioid prohormone proopiomelanocortin (POMC) gives rise to the intensified synthesis of antinociceptive endorphins, but also yields alpha-melanocyte-stimulating hormone (α-MSH) that promotes allodynia (pronociceptive effect) through its agonist action on melanocortin type 4 (MC4) receptors [28].

In parallel, agonist-induced stimulation of opioid receptors leads to their phosphorylation and reduced coupling to G proteins, hence decreasing their functionality [29]. This results in epigenetic silencing of Oprm1 and a subsequent downregulation of opioid mRNA and proteins in dorsal root ganglia (DRG) and spinal cord, as demonstrated in numerous rodent and primate NP models [30,31,32,33,34]. In human NP patients that suffer either from central or peripheral neuropathic pain, decreases in opioid receptor binding are observed in supraspinal regions: posterior midbrain, medial thalamus and the insular, temporal and prefrontal cortices [35,36]. The reduced supraspinal opioid receptor availability is in all likelihood a direct result of chronic pain itself (and not, for example, pre-existing individual differences), as similar decreased supraspinal MOP receptor expression is observed in animal models of NP [21,37]. The effect is accompanied by altered efficacy of G-protein stimulation within opioid-sensitive CNS structures, and altered expression of DOP and KOP receptor mRNAs, which suggests that neuropathy recruits and modifies several opioidergic circuits [38]. 

It is besides recognized that centrally expressed MOP receptors are more susceptible to downregulation after nerve tissue lesions [32,33,39] than the MOPs expressed on peripheral afferents. Coherently, central NP, resulting from lesions to brain and/or spinal cord, is considered to be less responsive to opioids than C, Aβ, and Aδ afferent fibers affecting peripheral injury-induced NP [40]. Apart from MOP receptor downregulation, nerve injury decreases their coupling to the voltage-gated Ca^2+^ (CaV) channels in primary afferent terminals, the process most likely mediated by a Src family kinase [41]. This in turn, uniquely for neuropathic pain, impairs the ability of MOP receptors to inhibit the release of pronociceptive neuropeptides, such as substance P [42].

To counteract endogenous opioids release-induced analgesia as to adaptively restore the nociceptive transmission balance, some neuronal hyperexcitability-promoting receptors are centrally upregulated under NP conditions, contributing to painful sensations. In rodent models of NP, neurokinin-1 (NK1) receptor activated by substance P, as well as antiopioid cholecystokinin (CCK) receptor, are upregulated in the spinal cord [43] and injured nerve [44]. The serotonin 2A (5-HT(2A), *N*-methyl-D-aspartate (NMDA) and mGlu1 (metabotropic) glutamate receptors augment the C-fiber evoked potentials and facilitate spinal nociception [45]. In addition, the upregulation of mGlu1 receptor contributes to the chronification of neuropathic pain as it underlies the increased the excitability of anterior cingulate cortex (ACC) layer 2/3 neurons, important for pain processing [46]. Other neuroplastic changes observed in painful neuropathy states involve the upregulation of bradykinin B1 and B2 receptors in the injured nerve, contributing to sensory nociceptive changes resulting in hyperalgesia [47]. 

### 1.4. Neuropathic Pain and Repeated Opioid Treatment Share Neuropathological Consequences

Remarkably, chronic opioid exposure is followed by spinal neuroplastic changes that overlap with the biochemical effects caused by nerve tissue injury, as both phenomena are associated with the activation of opioid receptors, induced either by exogenous opiates or the enhanced release of endogenous opioid peptides. As a consequence, repeated spinal delivery of a MOP receptor agonist, obviously relieving pain in acute administration, leads to the paradoxical development of allodynia and hyperalgesia, the painful sensations that are observed as well in the post-nerve-injury state. 

The commonality of these phenomena is reflected by the similar changes on the tissue level, which include opioid receptors downregulation in the CNS [48] and on the periphery [49] accompanied by the increased expression of pronociceptive neurotransmitters, such as CCK, calcitonin gene related peptide (CGRP) and substance P [50]. Similar increase is observed in case of glial activation which leads to neuroinflammation [51] and in the secretion of endogenous opioid peptides from leukocytes and macrophages via activation of leukocyte opioid receptors [52]. Besides, both conditions result in NMDA receptor activation and a subsequent initiation of intracellular cascades involving the increased activity of protein kinase C (PKC) and nitric oxide (NO)-activated poly(ADP-ribose) synthetase (PARS), contributing to behavioral manifestations of neuroplastic changes [53].

Having all the above listed evidence in mind, a question arises whether it is possible to eliminate or reduce the side effects of the opioid treatment while simultaneously addressing the opioid system-related pathological changes that are associated with neuropathic pain state, provided that certain alterations are common for the two conditions. This prerequisite can be met through designing drugs characterized by neuropathy-tailored hybrid structures, where the analgesic effect induced by the opioid pharmacophore is enhanced thanks to the simultaneous blocking of a rationally selected pronociceptive system, otherwise pathologically activated in NP (Scheme 1).

## 2. Targeting the Opioid System

Extensive research has been conducted to determine how the affinity to different opioid receptors influences the analgesia and other effects produced by exogenous opioid ligands, and if the desired therapeutic effects can be enhanced by multimodal targeting at the opioid receptors. Traditionally, MOP receptor activation is recognized to induce antinociception, and all clinically useful opioid analgesics are MOP receptor agonists [16]. On the other hand, high-affinity/high-selectivity MOP receptor agonists are burdened with pronounced side effects [54]. In this context, the multitargeting on opioid receptors is aimed to improve the drug’s tolerability and safety profile [55].

### 2.1. DOP Receptor Activation Alleviates the Neuroinflammatory Effects Induced by MOP Receptor Agonists

Results obtained in basic studies show that simultaneous activation of MOP and DOP receptors with bivalent agonist ligands delivers potent analgesia in experimental neuropathies [56,57] with strongly reduced dependence and tolerance development on the course of repeated administration [58]. The effect is explained by a reduced release of proinflammatory cytokines, such as TNF-α, provided by the repeated activation of DOP receptor [59] which in parallel promotes defense mechanisms acting against neuropathy-induced oxidative stress [60]. The increased glial-derived proinflammatory cytokine production is observed both in nerve injury and morphine tolerance states [51], so bifunctional MOP/DOP receptor agonist ligands can provide dual benefits as compared to single-target MOP receptor agonists, the latter expected to exacerbate gliosis and neuroinflammation that are already present in NP. Promising results have been obtained for biphalin, a dimeric enkephalin analogue that induces analgesia in NP via activating MOP and DOP receptors, while diminishing the expression of pronociceptive neuroinflammatory mediators released by activated microglia in vitro [61,62].

On the other hand, mixed MOP receptor agonist/DOP receptor antagonist nonpeptide ligands have low potential to induce tolerance and physical dependence [63,64], but their analgesic potency, superior to morphine, is demonstrated mainly in acute and chronic pain without neuropathic component [55].

### 2.2. Dual MOP/KOP Agonism in the Peripheral Nervous System (PNS) Attenuates Mechanical Hypersensitivity Symptoms 

Compounds displaying intrinsic efficacy at KOP receptors are found useful in specific subtypes of painful sensations in peripheral neuropathies: for example, in a mouse model of peripheral neuropathy induced by sciatic nerve ligation, a subcutaneous administration of KOP receptor agonist U-50,488H, but not morphine, alleviated dynamic allodynia, i.e., hypersensitivity to normally innocuous stimulus of gently stroking the affected skin with a cotton bud [65]. Such an observation is promising from the clinical perspective, as mechanical hypersensitivity is more common in NP patients than thermal hyperalgesias are [66], and dynamic mechanical allodynia interferes extensively with everyday life activities—it may be evoked merely by clothes touching the skin [67]. 

In the brain, KOP receptor activation opposes some MOP-mediated effects, such as tolerance and abuse liability-related reward processes [68], but it brings adverse events (dysphoria, sedation, diuresis) by itself, and contributes to the aversive aspects of ongoing pain [69]. On the other hand, peripheral KOP signaling provides some desirable effects in the nerve injury state, as it inhibits neuroinflammation and thus indirectly reduces nociceptor sensitization induced otherwise by inflammatory factors [70]. Activation of the peripheral KOP and MOP receptors with multifunctional ligands leads to efficient antinociception in peripheral painful neuropathies, while the side effects typical for MOP receptor activation are reduced [71]. Hence, recent drug development focuses on peripherally restricted KOP receptor agonists [72], and this approach appears useful also in the case of mixed MOP/KOP receptor agonists.

## 3. Rational Design of Hybrid Drugs

Once the pharmacokinetic profile of the opioid ligand aiming to treat neuropathic pain is tuned, another active pharmacophore can be added to the compound’s structure to provide further therapeutic benefits. Here is where the rational design of a hybrid drug begins (see Scheme 1). 

### 3.1. Dually Acting Hybrid Molecules Overcome the Limitations of Polypharmacy

The idea of hybrid drugs is underpinned by the general theory that biological systems are characterized by network structure, i.e., they are comprised of pathways and nods that influence each other to a different extent. As a consequence, the functioning of a biological system is supposed to be critically dependent on a few highly connected hubs, the disturbance of which results in a disease [73]. As abstract as it seems in the view of complexity of neuropathic pain mechanisms, it is actually observed that a combination therapy using two monofunctional agents may offer greater pain relief to the patients, as some drug sets provide synergistic effects [6]. Coherently, clinical polypharmacy for neuropathic pain is widespread, although the limited number of methodologically rigorous studies precludes the objective identification of a specific combination that would be superior to the others in terms of safety and efficacy [74]. 

Hybrid molecules are designed to overcome the limitations of polypharmacy, such as side effects overlap, poorly predictable distribution of distinct molecules in the tissue (the uneven distribution might hamper the desired effects), unwanted and/or unpredictable pharmacodynamic interactions, different pharmacokinetic profiles of the drugs delivered that preclude their additive or synergistic action, et cetera [75]. These benefits cannot be provided by drug cocktails or so-called multicomponent drugs (comprising two separate pharmacologically active agents packed into a single tablet) [76]. Instead, two overlapping pharmacophores, or separate moieties connected with a chemical linker, are composed into a single molecule that is able to aim at least two distinct molecular targets, as the pharmacophores act at their respective separate receptor monomers. In corresponding sections we specify that the discussed receptors, selected as hybrid compound targets, colocalize with opioid receptors in pain pathway structures, the latter implicating mutual interactions. As forming dimers and heteromers by opioid receptors and other GTP-binding protein (G protein)-coupled receptors (GPCRs) in vivo remains a controversial topic, and synthesizing a hybrid molecule that binds to a receptor dimer poses some technical difficulties (e.g., related to linker’s length and flexibility), designing a hybrid molecule that binds to only one receptor (targeted by one of its pharmacophores) at a time remains the most common approach. However, as such a molecule presents the affinity to at least two receptors, it is able to activate both types of receptors present in a tissue in a simultaneous manner, thus maximizing the biological effect. This pharmacodynamic characteristics of a hybrid compound helps avoid the scenario of hampered effect caused by uneven tissue distribution that could arise in case of two separate pharmacophores administered in a physical mixture. 

### 3.2. Hybrid Drugs Can Bring Specific Therapeutic Benefits in the Context of Neuropathic Pain Complexity

Apart from the technical gains listed above, the designed multiple ligands approach can bring further benefits that are specific for neuropathic pain. The complexity of NP as a pathology, let alone the multiplicity of its etiologies, encourages the search for multitarget therapeutic approaches to fight the symptoms efficiently. The molecular hybridization is a strategy that allows to improve therapeutic outcomes of NP treatment thanks to dual action that is meant to modify the activity of at least two pain processing-related systems at once. This way, each stimulation of an inhibitory (analgesia-producing) system is accompanied by a simultaneous blockade of a rationally chosen system that is known to counteract the inhibitory effects provided by the analgesic system under nerve tissue injury conditions. As the pain-promoting factors are blocked, the analgesic effects elicited by the antinociceptive pharmacophore can become more pronounced. The simultaneousness of reaching two distinct targets is crucial here, as it determines the final outcome of pain signal processing.

Designed multiple ligands are hoped to be the optional way of adjusting opioid agonists to make them better suited for neuropathic pain specifics, as the mechanisms of their reduced effectiveness and adverse effects observed in NP therapy are continuously better understood. Consequently, specific molecular targets can be rationally selected to apply the principle of modern hybrid synthesis to improve the opioid profile in a multiple administration model under nerve injury conditions. The following chapters discuss the systemic targets of hybrid molecules that are currently considered to be potentially useful in NP therapy. The general construction of a hybrid molecule designed to treat neuropathic pain is depicted on Scheme 2.

## 4. Neuropathy-Involved Systems as Targets of Hybrid Drugs

### 4.1. Opioid-Related Systems

#### 4.1.1. Melanocortin and the Melanocortin Type 4 (MC4) Receptor

Melanocortin receptors are five closely related seven-transmembrane GPCRs that are engaged in a vast variety of physiological processes, with key functions including pigmentation (MC1 receptor), steroidogenesis (MC2), energy homeostasis (MC3, MC4), immune reaction (MC5) and many more [77,78]. Their activity is regulated by peptide hormones (melanocyte stimulating hormones: α-, β- and, γ-MSH, and a larger peptide adrenocorticotropin—ACTH), derived by post-translational processing of an opioid prohormone named proopiomelanocortin (POMC) [79]. 

POMC belongs to the opioid gene family and incorporates both opioid and melanocortin core sequence, being able to be processed to an opioid peptide beta-endorphin and MC receptor-activating α-MSH [80]. It is of note that patients with chronic neuropathic pain have low levels of beta-endorphin in the CSF [81], while the level of POMC remains unchanged as compared to healthy individuals [82], which indicates that the processing of POMC can be altered in nerve injury state to produce pain-promoting effects. Nevertheless, the problem requires further studies to elucidate which factors, if any, are potentially able to influence the POMC-processing mechanisms.

##### Melanocortin System Exerts Tonic Pronociceptive Activity in Nerve Injury States

The expression of MC receptors is tissue-specific, and two of the five subtypes, MC3 and MC4 receptors, are the most abundant subtypes in the CNS and thus referred to as ‘neural’ melanocortin receptors [83]. Still, the involvement in pain regulation appears to be the function of the MC4 receptor exclusively, as its neuroanatomical distribution encompasses pain-processing regions of the peripheral and central nervous system. Coherently, many lines of experimental evidence point to the role of MC4 receptor in painful nerve injury states. There is evidence that MC4 receptor exerts tonic pronociceptive activity in painful nerve injury states, as blocking the receptor with exogenous antagonists alleviates pain-like behavior in rodent models of neuropathic pain [84,85] and reduces the markers of spinal neuroinflammation [86]. Other facts prove its influence on the functioning of the opioid system: blocking the MOP receptor increases the proallodynic effects induced by the activation of MC4 receptor [87], while selective MC4 receptor antagonists enhance the analgesic effectiveness of morphine [88] and prevent the development of morphine tolerance [89]. 

##### MOP/DOP Receptor Agonist-MC4 Receptor Antagonist Hybrids Provide Long-Lasting Analgesic Action in Preclinical NP Studies

All of these observations, as well as the overlapping distribution of MC4 receptor and opioid receptors in the pain-processing regions of CNS [90], inspired the development of opioid-MC4 receptor hybrid drugs. A significant contribution in this field was made by the group of Hruby, through their extensive research on the design of selective MC receptor ligands that resulted in the synthesis of SHU9119, an MC3 and MC4 receptor antagonist [91]. The finding preceded the development of the opioid (OP)-MC4 receptor bifunctional compounds [92] (however, their biological activity has not been determined) as well as trivalent opioid-melanocortin-cholecystokinin (CCK) ligands binding to the MOP, DOP, CCK1/2, and MC4 receptors [93]. SHU9119 was utilized to synthesize MOP/DOP-MC4 receptor binding hybrids with linkers of different length and rigidity. The preclinical tests in rodent models of NP performed by our group allowed to select the compounds characterized by the best analgesic profile under nerve injury conditions. Further authorial modifications to the compounds’ structures, based on the preclinical evidence obtained, gave rise to the series of bifunctional ligands uniquely designed to treat neuropathic pain [94,95]. The basic studies on mice and rats subjected to sciatic nerve injury (CCI model) revealed that the linker type prominently influences the compound’s properties in vivo, determining the intensity and duration of the final analgesic effect. All in all, the most effective OP-MC4 receptor hybrids occurred to reverse hypersensitivity-like behavior of CCI-subjected mice in i.t. doses about hundred times lower than individual pharmacophores. The effect provided by single-target parent compounds, as well as their physical mix, was less pronounced and relatively short-lasting (Scheme 3) [94,95]. The unique property of the OP-MC4 receptor hybrid, as compared to other ideas of bivalent compounds proposed for neuropathic pain treatment, is that its analgesic functions result from the modulation of a single nociception-related system, as the hybrid simultaneously enhances and suppresses the effects of different products derived from the same prohormone. 

#### 4.1.2. Nociceptin and the Nociceptin/Orphanin FQ (NOP) Receptor

A seven transmembrane-spanning GPCR, nociceptin/orphanin FQ (NOP) receptor is actually a member of the opioid receptors family and commonly classified as the fourth opioid receptor, as it is characterized by close homology to MOP, DOP and KOP receptors. After NOP receptor cloning, its endogenous ligand was unknown, and on this basis NOP was referred to as an ‘orphan’ receptor. Once identified independently by two groups, the endogenous heptadecapeptide resembling dynorphin A9–17 and presenting selectivity to NOP receptor was named ‘orphanin FQ’ (the two capital letters referring to the first and last amino acid in its sequence) by Reinscheid and colleagues [96] while Meunier et al. emphasized the pronociceptive properties of the peptide [97]. Eventually, a name, ‘nociceptin/orphanin FQ’ (N/OFQ), was coined.

Despite its high structural homology to opioid receptors, NOP receptor is insensitive to nonselective opioid receptor antagonist, naloxone, and it is not activated by endogenous opioid peptides [17]. The last piece of information allows to infer that the NOP system will not be affected by the same neuroplastic changes that can negatively influence the sensitivity of other opioid receptors to exogenous ligands, namely: resulting from the sustained endogenous opioids release that directly follows a nerve injury. This fact alone makes the NOP receptor an interesting target for hybrid drugs with opioid component designed to treat neuropathic pain. Still, further evidence makes NOP an attractive drug target in NP. 

##### Nociceptin System Induces Bimodal Effects on Pain Transmission

As mentioned above, early experiments with nociceptin showed that its supraspinal (i.c.v.) administration lowers the pain threshold in naive mice. However, later on, it was revealed that NOP receptor activation can bring either pronociceptive or analgesic effects, depending on the pain modality, ligand dosing, administration route, or animal species [98]. Nociceptin itself is able to induce spinal analgesia in nanomolar intrathecal doses, but, paradoxically, it lowers the pain threshold in naive mice when administered in lower, femtomolar doses, possibly by enhancing the pronociceptive substance P release [99]. Although NOP receptor agonists are able to exert spinal inhibitory effects in acute pain models [100], they appear to be the most effective as analgesics under nerve injury conditions [101,102] which suggests that the NOP system undergoes some neuroplastic changes as a consequence of nerve tissue injury. Intrathecal coadministration of nociceptin and morphine provides synergistic antihyperalgesic effects in rats [101] and nonhuman primates [103] but morphine antinociception can be potentiated by NOP receptor antagonists as well [104]. All in all, in the last decades it became clear that the nociceptin system can influence the pain transmission in a dual manner, and that both agonists and antagonists of the NOP receptor can modulate opioid analgesia. These observations inspired the development of mixed MOP/NOP receptors targeting hybrid drugs, as well as multivalent ligands possessing affinity to all four (NOP included) opioid receptors. 

##### MOP/NOP Receptors Targeting Hybrid Drugs are Safer than Selective Opioids

Preclinical studies with hybrid molecules targeting the NOP and other opioid receptors have been bringing promising results in the last few years. It is known that a clinically useful drug, buprenorphine, is a potent opioid analgesic acting as an agonist on the NOP, MOP and DOP receptors, while being an antagonist at the KOP receptor [105]. Still, the analgesic action of buprenorphine is apparently mediated primarily by the MOP receptor [106]. Hence, novel molecules with different intrinsic efficacy to the NOP (both agonists and antagonists are developed), as well as varying affinity to the three classical opioid receptors, are tested in animal models of acute, inflammatory and neuropathic pain. In particular, mixed MOP and NOP receptor agonist molecules are developed to obtain pharmacological agents displaying the highest potency and longest-lasting action in vivo [107]. 

In animal studies it is demonstrated that bifunctional MOP/NOP receptor agents alleviate nerve-injury induced hypersensitivity more efficiently than selective MOP or NOP receptor agonists, and produce long-lasting antinociception after spinal and systemic delivery in rodents and nonhuman primates [106,108]. Hybrid agents with partial agonist activity at MOP and NOP receptors are reported to display even wider therapeutic window than full MOP/NOP agonists, which allows to obtain a satisfying analgesic effect with lesser risk of adverse events. Coherently, bifunctional MOP/NOP receptors partial agonists are demonstrated to induce potent long-lasting analgesia without compromising physiological functions, such as cardiovascular activities and respiration, and not to produce physical dependence as they lack reinforcing effects. In addition, their repeated use is not associated with the development of opioid-induced hyperalgesia [104,109] which sums up to an outstanding safety profile. On the other hand, peptide-based mixed opioid agonist/NOP antagonist molecules alleviate painful neuropathy symptoms in far lower intrathecal doses (thousandths of a nanomole) than single opioid or NOP-targeting units, which also proves the utility of such hybrid drugs as potential as therapeutics in neuropathic pain states [110,111].

The promising results in basic studies led to the development of a new pharmacologic agent. Cebranopadol or GRT-6005 (trans-6′-fluoro-4′,9′-dihydro-*N*,*N*-dimethyl-4-phenyl-spiro[cyclohexane-1,1′(3′H)-pyrano [3,4-b]indol]-4-amine) is a small molecule activating nociceptin and opioid receptors dually, exhibiting a strong potency in rodent models of neuropathic pain [112,113]. It is currently undergoing clinical trials with the aim to treat painful diabetic neuropathy. Cebranopadol is a full agonist at the MOP and NOP receptors. Will the partial agonist MOP/NOP bifunctional ligands be the next novel analgesics developed for neuropathic pain patients?

### 4.2. Non-Opioid Systems

#### 4.2.1. Cholecystokinin (CCK) and the Cholecystokinin Type 2 (CCK2) Receptor

CCK is a peptide present in the CNS in the regions involved in pain modulation, i.a. periaqueductal grey matter (PAG), belonging to the descending pain inhibitory pathway, rostral ventromedial medulla (RVM) modulating the spinal nociceptive transmission, and dorsal (sensory) root ganglia and spinal cord. Described at first as a gut hormone, it started to be associated with pain processing as soon as it was discovered in the CNS by Vanderhaeghen et al. [114]. It was then observed that opioid and CCK peptides, as well as their receptors, present the overlapping anatomical distribution in spinal and supraspinal regions of the CNS. Early behavioral experiments showed that systemical or perispinal injection of CCK attenuates morphine-induced analgesia [115], while the CCK-receptor antagonist, proglumide, potentiates the analgesic effects of morphine, as well as prevents the development of tolerance to these effects [116]. At present it is known that RVM neurons co-expressing MOP and cholecystokinin type 2 (CCK2) receptor facilitate pain and—together with CCK interneurons in the deep dorsal horn—contribute to the maintenance of tactile allodynia in chronic painful neuropathy [117,118]. These findings are indicative of the antiopioid action of the CCK system. 

##### Activating the Opioid Receptors along with Blocking the CCK Receptor Provides Synergistic Analgesic Effects

The antiopioid activity of the CCK receptor was further confirmed in studies utilizing targeted mutagenesis involving CCK receptor knock-out mice, and testing different specific antagonists of CCK receptors [119,120]. Importantly, co-administration of CCK antagonist and morphine provides synergistic, and not only additive, analgesic effects in a rat model of neuropathic pain [121]—the synergy creates an opportunity to lower the effective dose of an opioid agonist to avoid opioid-related side effects while obtaining robust analgesia. In the light of these findings, chimeric bifunctional opioid agonist/CCK receptor antagonist compounds were designed for the treatment of neuropathic pain [122]. The hybrids from this group revealed a promising analgesic profile with reduced side effects when tested in the spinal nerve ligation model of neuropathic pain [123].

#### 4.2.2. Substance P (SP) and the Neurokinin 1 (NK1) Receptor

Neuropeptide SP is one of the three products of Tac gene processing, next to neurokinin A and B. The three peptides constitute the tachykinin family and are broadly distributed in the organism, participating in vital physiological functions, such as immune, gastrointestinal and respiratory-related processes. There are three known subtypes of tachykinin receptors and SP can bind to all three of them, but it activates neurokinin-1 (NK1) receptor preferentially. 

11-amino acid SP is synthesized in dorsal root ganglia and released from primary afferent fibers as a result of noxious input to transmit nociceptive signals to spinal and brainstem second-order neurons [124,125] contributes to central sensitization in neuropathic pain states: it is known to be overly released in injured nerves [44] and to sensitize spinal neurons to input [126]. Coherently, it was observed that blocking the spinal NK1 receptor delays the onset of neuropathic pain [127]. In parallel, SP mediates neurogenic inflammatory responses via mast-cell-specific receptors, enhancing the release of proinflammatory cytokines and chemokines [128] that can further exacerbate neuropathic pain symptoms and oppose opioid analgesic action [129].

##### NK1 Receptor Blockade might Reduce Certain Opioid Use-Related Side Effects

The selective antagonism on NK1 receptors clearly modulates hypersensitivity-related behavior in animals subjected to nerve injury, alleviating most of the observable symptoms of such hypersensitivity [130], but it can bring broader benefits in terms of opioid treatment, as the receptor is involved in a handful of processes that can be disturbed by the repeated opioid use. Therefore, blocking the NK1 receptor is expected to prevent or attenuate some opioid-related adverse events.

Rewarding properties of opioids are a serious burden that can enhance the risk of misuse and opioid addiction. Importantly, NK1 receptor-containing neurons in the amygdala mediate acute opioid reward [131]. Besides, NK1 receptor appears to play a critical role in the motivational properties of opioids, modulating negative emotional states, such as anxiety, that are often associated with opiate withdrawal once the addiction establishes. NK1 receptor antagonists are shown to reduce opioid self-administration behavior and stress-induced drug seeking in rats [132], and as such might contribute to the improved safety profile of an opioid agonist once it is fused with an NK1 receptor antagonist to form a hybrid drug. 

As opioid receptors are abundant in the gastrointestinal tract, some common and burdensome adverse events of opiate treatment result from the inhibitory effects that exogenous opioids exert on the gastrointestinal functions. Due to decreased gastric emptying that opiate therapy might cause, about 20–30% of opioid-treated patients experience nausea or vomiting. In this context, the antiemetic properties of NK1 receptor antagonists may come in useful [133]. 

When it comes to the experimental evidence, analgesic doses of newly designed hybrids: opioid agonist-NK1 receptor antagonists are reported to actually produce less opioid-related adverse events, like constipation [134], sedation [135,136], reward liability [135] or quick development of tolerance to the antihyperalgesic effect [134,135] observed typically in monofunctional opioid agonists. No cross-tolerance effect with morphine observed after repeated use of opioid-neurokinin 1 (OP-NK1) receptor hybrid is another promising result, as it indicates that OP-NK1 receptor hybrids are applicable for treatments involving opioid rotation [137].

### 4.3. Emerging Directions

New ideas on the hybrid drugs designed for neuropathic pain treatment are continuously emerging. The following chapter presents some of the latest findings in the field, as well as current challenges that await the solution.

#### 4.3.1. Dynorphin/Bradykinin and the KOP/Bradykinin (B1/2) Receptors

The kinin system widely contributes to the sensory changes associated with neuropathic pain states; next to the already discussed NK1 receptor actions, kinin B1 (injury-induced) and B2 (constitutively expressed) receptors [138] play a vital role herein—importantly, in the course of interaction with the opioid system [139]. 

##### Opioid Neuropeptide Dynorphin Exerts Antiopioid Effects via B1/2 Receptors

It has already been mentioned that dynorphin, an inhibitory opioid neuropeptide derived from the precursor prodynorphin, is paradoxically able to promote pain and hypersensitivity via non-opioid mechanisms [140]. It was shown that pathological elevations of dynorphin may activate spinal B1/2 receptors to mediate and maintain neuropathic hyperalgesia [141]. The elevations in the spinal dynorphin content are seen not only in neuropathy, but also in conditions of opioid-induced pain states [142]. In parallel, the studies on rodent models of neuropathic pain reveal that the expression of the excitatory bradykinin receptors is enhanced in the CNS and PNS in painful neuropathy states [143]; therefore, once the its release is elevated, dynorphin can exert pronounced antiopioid effects through binding to bradykinin receptors, and the effects may be counteracted with B1 and B2 antagonists to alleviate the hypersensitivity [27,47]. In the view of these findings, blocking the bradykinin receptors appears as a rational strategy to potentiate the opioid-induced analgesia, otherwise hampered by the excitatory, bradykinin receptors-mediated actions of dynorphin.

Recently, Deekonda et al. proposed the design and synthesis of novel multifunctional ligands, where opioid enkephalin analogues were conjugated to Hoe 140, a ligand acting as a B2 receptor antagonist when tested in Guinea pig ileum. However, the group reported that the novel compounds show weak binding affinity for rat B2 receptors [144]. Hence, the idea of developing the OP-bradykinin receptor targeting hybrid compound still remains open. 

#### 4.3.2. Neurotensin and the Neurotensin (NTS) Receptor

Broadly distributed in the CNS, two main neurotensin (NTS1/2) receptors (GPCRs classified into two subtypes: high-affinity NTS1 and low-affinity NTS2 receptors) modulate nociception, pathological pain and stress-induced antinociception, exerting mostly inhibitory effects on neural transmission when activated [145]. The role of two additional neurotensin-binding non-GPCR receptors, Sortillin/NTS3 and sorLA/LR1 1, is less understood up to date. Neurotensin itself is a tridecapeptide involved in physio- and pathological processes of the CNS and gastrointestinal tract [146], exerting bipolar effects on pain transduction: facilitatory in low picomolar doses and inhibitory in higher nanomolar doses, which was shown by Smith et al. using i.c.v. microinjections in the rat [147]. Experimental evidence show that neurotensin is coreleased with SP in the rat spinal cord once nerve injury-induced allodynia is developed, probably balancing the excitatory effects of SP by inhibition of nociceptors [148]. 

##### OP-NT Receptor Hybrid is yet to be Tested in Neuropathic Pain Assay

The important characteristics of NTS1 and NTS2 receptor in the context of hybrid drug design is that they mediate analgesic actions via opioid-independent mechanisms [149]. As a result, BBB-crossing analogues of neurotensin used as analgesics possess favorable side effect profile devoid of some opioid-related adverse events [150], while tolerance to their analgesic effect does not develop on the course of repeated administration [151]. Coadministration of a brain-penetrant neurotensin analogue with morphine resulted in an additive analgesic effect in an inflammatory pain assay, but the effective dose of morphine could be kept low this way, thus allowing to reduce adverse events [152]. Promisingly, analgesic synergy of morphine and NTS2-selective agonist has been observed in a formalin-induced persistent pain model [153]. In neuropathic pain assays, NTS2 receptor agonists attenuate not only hypersensitivity-related behavioral responses triggered experimentally during the test procedure, but also reverse postural deficits acquired as a result of nerve damage, unlike morphine [154]. These experimental data imply that neurotensin analogues can reinforce the analgesic action of opioids in a complementary manner, providing some unique additional benefits in the dual treatment. Up to date, improved (as compared to a physical mixture of parent structures) analgesia/side effect ratio of an endomorphin-2/neurotensin combining (OP-NT) hybrid drug was demonstrated in rodent assays of acute pain [155]. The properties of precursory OP-NT hybrid compounds in neuropathic pain assays are yet to be tested.

#### 4.3.3. Glutamate and the Metabotropic Glutamate 5 (mGlu5) Receptor

Glutamate is the most abundant excitatory neurotransmitter in the nervous system, physiologically released i.a. by sensory afferents in the spinal cord’s dorsal horn. Nerve injury is followed by the excessive release of glutamate from primary afferents and spinal interneurons, which leads to the nociception-promoting plastic changes and consecutive hyperexcitability of both lesioned and intact nerve fibers located in their close neighborhood. The glutamate-driven hypersensitivity manifests itself in evoked and spontaneous painful sensations typical for chronic neuropathic pain [156]. To counteract the heightened excitability caused by the rise in extracellular glutamate, antagonists of metabotropic glutamate receptors can be utilized, with mGlu5 receptor antagonists shown to be analgesically effective in many neuropathic pain assays [157]; in particular, 2-Methyl-6-(phenylethynyl)pyridine hydrochloride (MPEP) has been widely tested in diverse neuropathic pain models, with promising results [158,159]. Next to its hypersensitivity-attenuating properties in NP assays, MPEP was shown to potentiate the analgesic effects of acutely administered morphine, while preventing the development of tolerance on the course of repeated morphine administration [160]. Developing a ligand acting dually at opioid and mGluR5 receptors constitutes a relevant continuation on these studies.

##### Certain Controversies on the Actions of OP-mGlu5 Hybrid Need to be Elucidated

A molecule that combines oxymorphamine (MOP agonist) and MPEP (mGlu5 receptor antagonist), MMG22, showing an exceptional analgesic potency in mouse inflammatory pain assay, was designed and synthesized by Akgün and colleagues [161]. The group hypothesized that the bivalent compound acts through MOP/mGlu5 receptors heteromers. However, this interpretation was later on questioned on the basis of the results of detailed experiments, though confirming the synergistic interactions of the two pharmacophores in neuropathic pain model [162]. The most recent study examined the adverse events potentially associated with MMG22 systemic administration, and showed a superior safety profile of dually acting MMG22 as compared to traditional opiate treatment [163]. The results reveal themselves as more than promising; on the other hand, Vincent and colleagues point out that only the blockade of intracellular mGlu5 receptor (mostly located on nuclear membranes) exerts the most pronounced molecular and systemic effects in the context of neuropathic pain-driven behavior [164].

## 5. Limitations and Challenges of the Hybrid Drugs Approach

The ultimate goal of pharmacological research is to translate a given new drug candidate into an effective therapy in the clinic. Although promising and increasingly successful thanks to the implementation of innovative in silico tools, the rational multitarget drug design approach is not devoid of tricky points. Selecting pharmacological targets that can bring desirable therapeutic outcomes when modulated simultaneously, as well as choosing high-quality basic (‘parent’) moieties to be incorporated into a hybrid molecule, is just the very first step of the process. Importantly here, all the biological characteristics of chosen targets should be carefully analyzed, to ensure that the antiopioid action of a selected system, meant to be counteracted by a novel hybrid, actually results from activating the respective pain-promoting receptors(and not from other mechanisms). For example, bradykinin receptors, targeted by dynorphins, are proposed as an adjuvant target of opioid-derived hybrid compounds, but there is evidence that pathological effects of dynorphins observed in vivo result from their ability to form pores in cell membranes [165]. Relevant experimental validation, e.g., using specific receptor antagonists, should then precede the hybrid compound design and synthesis.

While designing a multifunctional drug, once the target combinations are validated, the compound must be optimized in terms of structure-activity relationships so that its desirable drug-like properties could be retained in the combined molecule. It must not be ruled out that given pharmacophores would not be characterized by the same affinity and efficacy to their target receptors once welded together, not excluding that their activity at the receptor site is eventually opposite to the starting pharmacodynamics profile (e.g., an agonist retains its affinity to the receptor but loses its efficacy and becomes an antagonist). To start with, a careful in vitro analysis revealing the pharmacological characteristics of a newly synthesized hybrid particle, with the target receptor binding affinity and efficacy remaining the crucial parameters, is of great importance. The linker (i.e., spacer that connects two ligands) type is also a factor that influences the pharmacological activity of a hybrid drug to a great extent, as it contributes to the stability of the molecule. Experimental studies show that the linker’s length and rigidity may greatly enhance the analgesic potency and longevity of the therapeutic effect provided by a given compound, likely by decreasing its susceptibility to enzymatic degradation [161]. The next important quality of a hybrid drug, and a considerable limitation of the multifunctional compounds approach, is the size of the designed particle which eventually determines its bioavailability and possible routes of administration, as big molecules will not be able to effectively penetrate into the CNS.

The translational aspect remains perhaps one of the greatest challenges related to the drug development process. Preclinical studies provide vast amount of detailed data on synergistic analgesic effects of distinct ligands, demonstrated in numerous animal models of pain and hypersensitivity. Nevertheless, the evergreen question is whether the promising results obtained in animal models of neuropathic pain would bring comparable benefits in human subjects. Some strong candidates to boost opioid effectiveness backed up by substantial basis of preclinical data did not provide similar satisfying results in patients. For example, CCK2 receptor antagonist L-365,260 failed to augment the analgesic effect of morphine when applied as a co-treatment in human subjects with chronic neuropathic pain [166]. Similarly, NK1 receptor antagonists tested in clinical studies did not present analgesic properties comparable to those observed in rodent models [167]. To explain the discrepancies, it is pointed out that the drug-induced modulation of behavioral responses at a level detectable in animal tests is not equal to analgesia [168] and interpreting it as such should be considered a fallacy. Differences between species, not excluding the most subtle variations in genome organization and regulation [169], are another possible answer to the same question, and studies in nonhuman primates seem superior to rodent models in this context—still, most pain assays are excluded from testing on primates [170]. Besides, there are numerous factors to be considered when interpreting the results of preclinical studies, starting from the most general variables, such as sex and age of the animals, to the detailed specifics of a chosen pain assay that may highly influence the final results [171]. The immanent quality of behavioral tests is that they measure specific symptoms of neuropathy, and each of the symptoms (e.g., mechanical allodynia or thermal hyperalgesia) results from distinct systemic and molecular mechanisms. Only detailed studies reveal that, for example, SP participates in heat hyperalgesia, but it does not contribute to cold hypersensitivity or mechanical allodynia arising from nerve injury [172]. This, at least in part, can pose an answer on the question why blocking the NK1 receptor is not a versatile pharmacological tool to alleviate neuropathic pain symptoms. 

## 6. Summary

Opioid agonists in their prolonged action, although providing analgesia, steadily induce molecular and systemic changes that can exacerbate painful sensations in neuropathy and eventually outbalance the initial analgesic effects provided by opioid receptors activation. In parallel, opioid agonists in equivalent doses tend to be less effective in NP than in acute or inflammatory pain due to the specific antiopioid pain-promoting mechanisms that typically follow nerve tissue injury. The occurrence of these phenomena is inevitable unless the undesired effects, both resulting from the nerve tissue injury and the prolonged treatment with opioid analgesics, are pharmacologically silenced from the very beginning of the therapy. The continuously better understanding of the common features of neuropathic pain and opioid exposure creates a worthwhile opportunity to escape the additive antiopioid effects of the nerve injury and opiate treatment. 

As outlined in the present review, hybrid drugs approach creates some promising perspectives in this challenging attempt, as the multifunctional structures allow to enhance and prolong the opioid analgesia thanks to the simultaneous modulation of unwanted pronociceptive effects, exerted otherwise by a given neuropathy-activated system. What is more, some hybrid molecules might provide new beneficial quality as compared to the effects of their parental compounds; the prolonged analgesic action that grows up in time, observed in the case of a selected opioid agonist-MC4 receptor antagonist hybrid (see Scheme 3) exemplifies this phenomenon. 

The choice of the ancillary moiety to be incorporated into an opioid-derived hybrid compound remains a vitally important task, which once accomplished, while being backed up by well-grounded, high-quality experimental evidence delivers the possibility of designing an efficient and safe analgesic dedicated for neuropathic pain treatment.

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
