# Peer review of "Multifunctional Opioid-Derived Hybrids in Neuropathic Pain: Preclinical Evidence, Ideas and Challenges"

_molecules, 2020, doi:10.3390/molecules25235520_

Round 1

Reviewer 1 Report

In this paper the Authors report some new about opioid hybrid drugs designed to treat neuropathy, focusing on their biological action, reporting limitations and future develop related to their possible therapeutic use. Firstly, the Authors describe the main characteristics of neuropathic pain and the efficacy of current analgesic therapy and discuss the interactions between neuropathic pain mechanisms and the opioid system. Then the Authors briefly debate on the different role of opioid receptors in analgesia, the effects of multimodal targeting at the opioid receptors and the rationale design of hybrid drugs. After this introductory part, the Authors discuss i) the effects of hybrid drugs based on the opioid system such as the MOP/DOP Receptor Agonist-MC4 Receptor Antagonist Hybrids and MOP/NOP bifunctional ligands and ii) the effects of hybrid drugs based on mixed systems such as chimeric bifunctional opioid agonist/CCK receptor antagonist and  opioid agonist-NK1 receptor antagonist. Following, the latest findings and challenges on the hybrid drugs designed for neuropathic pain are discussed including the effects of hybrid as encephalin analogues (MOP agonist)/Hoe140 (B2 receptor antagonist), [Ile9]PK20  an opioid-neurotensin peptide and oxymorphamine (MOP agonist)/MPEP (mGlu5 receptor antagonist). Finally, the limitations and challenges of the hybrid drugs approach to treat neurophatic pain is discussed.

The Authors summarize in this review the current status of hybrid compounds potentially able to fight neuropathic pain. The article gives an interesting historical and scientific perspective on this field focusing on the most promising compounds. Interestingly, not only the effects of these hybrids are discussed but also limitations and challenges related to their possible therapeutic use are briefly discussed. I believe that this article can be a valid reference for those who work in the field.

Reviewer 2 Report

Paper by Starnowska-Sokół and Przewłocka1discusses the rational of hybrid drug designs for use in alleviating neuropathic pain. The authors focused on opioid drugs for treating neuropathic pain. I have some comments on this review:

1-Title: is not perfect to outline the content of the review, for example, authors did not mention any info regarding that they are talking about opioid drugs mainly

2-Abstract: is not accurate and gives impression to the reader that opioids are the first or only line treatment of neuropathic pain and this is not correct & also there is a bias in writing towards opioids and this needs revision to be more realistic. The authors classified the medications in a better way in INTRoduction.

I wish if the authors revise this statement or otherwise provide recent references :

Patients with neuropathic pain (NP) are prescribed opioids more often than individuals affected by chronic pain of non-neuropathic origin. 

3- The number of references is too high, needs reduction to max of 100.

4- Some parts of the review need to follow the general aim to give a straight-forward impression to the reader
In section 4.2. : do the authors feel this section matches other parts ?
I wish if the authors focus on the primary target.

Reviewer 3 Report

Joanna Starnowska-Sokół and Barbara Przewłocka in “Multifunctional Hybrids in Neuropathic Pain: Preclinical Evidence, Ideas and Challenges” review how hybrid molecules may overcome the limitations of polypharmacy.

This is well-written and comprehensive paper covering most aspect of this emerging field. I think that it may be accepted for publication as it is or with minor modifications.

However, what is still unclear to me is:

  1. whether there is any example of hybrid molecule that blocks neuropathic pain qualitatively better than its two constituents separately;
  2. whether there is any example of hybrid molecule that displays quality that is different from any feature of two parental compounds;
  3. because molecular targets may be located in different areas or even tissues, whether pharmacodynamics and pharmacokinetics of hybrid molecules is more or less beneficial vs. those of two individual drugs.

Please add this information, or emphasize ideas on these issues in the respective parts of the review.

Specific comment: dynorphins are sticky and may bind in vitro to many protein molecules, even to BSA. Dynorphin non-opioid actions in vivo and in vitro may be due to their ability to induce giant stochastic pores in plasma membrane. Poration by dynorphins may represent a mechanism of pathological signal transduction (see please, Maximyuk et al., Cell Death Dis. 2015, 12; 6(3):e1683. doi: 10.1038/cddis.2015.39). Thus interaction with B1/2 and other receptors may be either in vitro artifact, or secondary to pore formation.

Therefore, these molecules probably are not a good case for design of hybrid molecules. However, because their ability to translocate across the plasma membrane they, similarly with penetratins, may be used for delivery of cargo into the cell.

Round 2

Reviewer 2 Report

I recommend acceptance of the current form